# Scanning for Obesogenicity of Primary School Environments in Tshwane, Gauteng, South Africa

**DOI:** 10.3390/ijerph20196889

**Published:** 2023-10-06

**Authors:** Morentho Cornelia Phetla, Linda Skaal

**Affiliations:** 1Department of Human Nutrition & Dietetics, Sefako Makgatho Health Sciences University, Pretoria 0001, South Africa; 2Faculty of Health Sciences, Durban University of Technology, Durban 4001, South Africa; lindas@dut.ac.za

**Keywords:** primary schoolchildren, obesity, obesogenic environment, ANGELO framework

## Abstract

The purpose of this study was to scan for the obesogenicity of primary school environments in Tshwane, Gauteng, South Africa. This study was carried out in ten public primary schools in the Tshwane West district of the Gauteng province. An observational design was used to collect the data. Data collection was conducted using an observational checklist guided by the Analysis Grid for Environments Linked to Obesity (ANGELO) framework. The findings revealed that although a nutrition policy was available in most schools, few schools were communicating the policy. Despite all the schools having sports fields, children were not exercising. Most schools also had a school nutrition programme. Most primary schoolchildren’s lunchboxes contained sweets, sugar-sweetened beverages, and snacks. A few lunchboxes contained fruits and vegetables. Primary schools in Tshwane West did not comply with most aspects of the school mapping environment, indicating that the school environments were obesogenic. It is therefore essential to scan for obesogenicity in all South African schools so that tailor-made interventions can be implemented to rectify and further prevent obesogenic school environments.

## 1. Introduction

Childhood obesity is a global public health issue, affecting more than 340 million children and adolescents between the ages of 5 and 19 years worldwide [1]. The World Health Organisation (WHO) [2] predicts that by the year 2025, there will be approximately 70 million overweight or obese children under the age of five. Globally, there has been a steady increase in childhood obesity, from 6.7% in 2010 to 9.1% in 2020, and the numbers are increasing annually [1].

Childhood obesity affects children from both developed and developing countries. Studies show a high prevalence of childhood obesity in developed countries in Europe [3,4] and the United States of America (USA), where 20% of children are obese. Developing countries record similar trends to those of developed countries, with childhood obesity prevalence rates of 17.4%; 18.5%; 20.2%, and 22.6% in Nigeria, South Africa, Ethiopia, and Tanzania, respectively [3,4,5]. Obesity is prevalent among all age groups in South Africa, with childhood obesity ranging from 14.2% to 18.5% [2,6]. 

Reports indicate that the risk of obesity starts in the first year of life, stabilises, and then peaks at the age of 6 years, where girls are reported to be more obese than boys [2,7]. The 2016 Strategy for the Prevention and Control of Obesity in South Africa [6] reports that 14.2% of primary schoolchildren are overweight and obese, with the Gauteng province reporting 5.2% more obese girls than boys.

According to Rendina, Campanozzi and De Filippo (2019), an obesogenic environment is an environment that predisposes children to obesity through the promotion of a sedentary lifestyle [8]. Documented factors which contribute to obesity among schoolchildren include environmental factors, lifestyle preferences, cultural environments, increased caloric and fat intake, lack of physical activities, a preference for fast foods, frequent snacking, increased portion sizes, and the increased intake of sugar-containing beverages [9,10]. Globalisation has negatively impacted the nutrition of most South Africans, especially those living in urban areas. The nutritional transition means that the modern diet consists of high fat, high sugar, low dietary fibre, and processed foods [11]. Children are exposed to fast food outlets very early in life as a result of aggressive marketing and easy accessibility, and school outlets are complicit in selling these unhealthy foods [12,13]. Evidence shows that South African schoolchildren are exposed to obesogenic school and community environments, which enable a sedentary lifestyle by promoting physical inactivity and poor eating habits while failing to provide a conducive environment for schoolchildren to buy and eat healthy foods or take part in exercise [14]. Furthermore, schools are surrounded by tuck shops, street vendors, and retailers that sell unhealthy foods The Tuckshop is a ‘traditional’ store on school premises that sells confectionary snacks and soft drinks [15]. The increased supply of ultra-processed high-energy foods and the lack of physical activity are key drivers of obesity worldwide [1].

In a bid to eradicate food insecurity for schoolchildren, the Department of Education implemented the National School Nutrition Programme (NSNP) in primary and secondary schools [12]. As part of that, the NSNP Guidelines for Tuck Shop Operators (2014) were designed by the South African Department of Basic Education (DBE). These guidelines were intended to inform tuck shop operators of the types of healthy foods that can be sold to schoolchildren and which unhealthy foods to avoid. However, there is no standardised monitoring of the operations of tuck shops and street vendors to ensure that they comply with the set guidelines. Furthermore, documented challenges with the implementation of NSNP range from food suppliers’ non-compliance, improperly prepared food, and menus not including fruits and vegetables as recommended by South African Food Based Dietary Guidelines (SAFBDG) [16].

Studies also show that poor nutrition and physical inactivity contribute to the obesogenicity of the school environment [9], despite evidence showing that sport and physical activity could be effective in preventing obesity [17]. Unfortunately, school periods allocated for physical education/activities are often used for other academic activities or become free periods for learners to idle outside the classrooms. Other documented challenges relate to the inability of schoolteachers to incorporate actual physical activity into the Annual Teaching Plan (ATP), further contributing to physical inactivity. In some cases, poorly maintained or absent sports equipment and facilities have been found to contribute to poor sports quality and frequency in schools [18].

Ideally, school environments should promote a culture of healthy eating and enable physical activity as a contribution to the prevention of childhood obesity [19]. There is a need for more research examining the obesogenicity of the school environment in order to inform the Department of Basic Education about the additional resources needed to improve school environments and prevent childhood obesity among children. Despite government intervention, it has been documented that the rates of obesity among primary schoolchildren in South Africa continue to rise. Could this be due to the obesogenicity of school environments? The aim of this study was to scan the obesogenicity of primary school environments in Tshwane, a metropolitan municipality in the Gauteng province of South Africa using the Analysis Grid for Environments Linked to Obesity (ANGELO). The hypothesis of this study was that Tshwane primary school environments are obesogenic.

## 2. Materials and Methods

This study used a cross-sectional descriptive design, which analyses variables collected at one point in time across a population. Data were collected using a checklist and observation. The total number of schools eligible for inclusion in the study was 31. The eligibility criteria were based on the researcher’s accessibility quintile levels. A multistage stratified sampling technique was applied to sample ten public primary schools in the Tshwane West district of Gauteng. Stratified random sampling was used to sample primary schools in the context of quintile levels. The stratification of primary schools was as follows: The names of the primary schools in all areas were categorised in the context of five quintiles, and two schools were randomly selected from each quintile, making up the ten schools that participated in this study. In South Africa, quintile 1 schools are classified as the poorest schools, quintile 2 schools are classified as the next poorest, and quintile 5 schools are classified as affluent schools. In this study, schools that fell into quintiles 1 and 2 were grouped and classified as low socioeconomic schools, quintiles 3 and 4 were classified as middle-class schools and quintile 5 as affluent schools.

The researchers used the Analysis Grid for Environments Linked to Obesity (ANGELO) framework [20], which is a tool that assesses all the societal and environmental drivers of obesity in four domains, namely the physical, economic, legislative, and sociocultural domains. The ANGELO framework was used as a guide for the development of the observation checklist used in this study, as outlined in Figure 1. Data were collected through observation and a ticking checklist. The data were collected by the researcher and research assistants who spent prolonged periods at each school. Observations were conducted on a random basis for three months. Areas that were observed included the kitchens, classrooms, sports grounds, play areas, tuck shops, street vendors, and inside and outside the surrounding areas of the schools. Lunchboxes were inspected during lunch breaks. Schoolchildren were observed while eating during breaks. Life Orientation (L.O.) educators were observed during teaching in order to establish the depth of the content covered for both physical education and nutrition education. Questions on the checklist regarding the availability and implementation of nutrition and physical activity policies were posed to school principals as these cannot easily be observed.

The checklist for the physical domain involved the observation of school feeding programmes, nutrition and exercise education, sports fields or play areas, and exercise activities. The economic domain involved the observation of street vendors and tuck shops to see the types of food being sold. The legislative checklist involved the observation of the NSNP and the Department of Basic Education (DBE)’s Guidelines for Tuck Shop Operators. The sociocultural checklist involved the observation of the availability of gardens, the types of food being consumed from the school feeding programme, and the contents of lunchboxes. A total of 250 lunchboxes were randomly inspected in the ten schools. The contents of the lunchboxes were classified as snacks (Simba chips, biscuits, etc.), sweets (lollipops, chocolates, etc.), junk foods (bunny chow, potato chips, pizzas, etc.), sugar-sweetened beverages (SSBs) (cool drinks, sweetened drinks), vegetables (cucumber, lettuce, carrots, etc.) and fruits (raw fruits, fruit salads, 100% fruit juice, etc.). The checklist was developed to assess the contents of the lunchboxes. The presence of an item in the lunchbox was marked with a tick.

### 2.1. Ethical Considerations

The ethics considered in this research were seeking permission, informed consent, confidentiality, privacy and anonymity, nonmaleficence, beneficence and autonomy, and justice. This study obtained ethical clearance from the Turfloop Research and Ethics Committee. The study project was assigned the following reference number: TREC/343/2019: PG. Written permission to conduct this study was obtained from the National Department of Basic Education and the Tshwane West District Office. The principals of all ten primary schools gave verbal permission. Written parental consent was sought from the parents. The schoolchildren were also required to sign assent forms before data collection. Confidentiality and privacy were guaranteed.

### 2.2. Data Analysis

Data were coded and entered into the Statistical Package for the Social Sciences (SPSS) version 27 for analysis, where descriptive statistics were used to establish frequency distributions.

## 3. Results

Figure 1 below shows the observation checklist used to assess all the societal and environmental drivers of obesity in four domains, namely the physical, economic, legislative, and sociocultural domains.

Table 1 below shows that of the ten sampled schools, the majority (90%) had a nutrition policy, and 40% communicated the policy to all staff. Only 20% of the schools had nutrition posters, while 100% of schools had street vendors in their vicinity. Only half (50%) had school tuck shops and none of the schools implemented the DBE Guidelines for Tuck Shop Operators. Only one school had a copy of the South African Food-Based Dietary Guidelines (SAFBGs). Two schools (20%) had vegetable garden infrastructure, albeit nonfunctional.

Table 2 below shows that despite 100% of the schools having dedicated sports fields, none of the primary schoolchildren exercised (note: this study was conducted during COVID lockdown stages). No exercise posters or brochures were available despite all the schools having exercise education as part of their curriculums. Half of the schools (50%) were fully equipped with exercise equipment; however, only one school had experienced exercise instructors for exercises and was able to design exercises according to the DBE’s Annual Teaching Plan (ATP).

Table 3 below shows that the majority of schools (*n* = 9) had a school feeding programme; half (50%) followed the prescribed menu and had adequate food service units (FSUs) and 80% of the children finished the food they received from the feeding programme.

Figure 2 below shows the contents of lunchboxes according to the socioeconomic status (SES) of the school. Approximately 250 lunchboxes of 250 children from different quintiles were inspected. Of these, most lunchboxes contained sugar-sweetened beverages (SSBs) (93%), and very few contained fruits (25%) and vegetables (1%). The lunchboxes of children attending middle-class schools mostly contained sweets (97%) and very few had fruits (33%) and vegetables (5%). Most of the affluent schoolchildren had snacks (96%) and very few (6%) had vegetables.

Table 4 shows that there was no association between the contents of the lunchbox according to snacks, fruits and vegetables, however, the fluent lunchboxes contained more fruits, vegetables and snacks. Furthermore, there was an association between SES in terms of junk (*p* < 0.01, 95% CI), sugar-sweetened beverages (*p* < 0.001, 95% CI), and sweets (*p* = 0.07, 95% CI); the lunchboxes of learners from middle- and low-socioeconomic schools that contained more junk foods, sugar-sweetened beverages, and sweets, as compared to those in affluent schools.

Table 5 below illustrates the post hoc test for lunchbox contents vs. school-level SES. The adjusted *p*-value indicates snack association between the consumption of snacks and low socioeconomic status (*p* = 0.0014, 95% CI). There is a true association between junk food among middle SES (*p* = 0.0061, 95% CI) and affluent SES (*p* = 0.00693, 95% CI).

There is a true association between the consumption of fruits and SES; there was higher fruit consumption among those with affluent SES (*p* = 0.000, 95% CI) and lower fruit consumption among those with low SES (*p* = 0.001, 95% CI). Also, the adjusted *p*-value indicates that there is a true association between vegetable consumption in terms of SES, there is a high consumption among those with affluent SES (*p* = 0.000, 95% CI) and low consumption among the low SES (*p* = 0.000, 95% CI). The adjusted *p*-value indicates that there is a true association between sweetened beverage consumption and low SES (0.0019, 95% CI) but not with middle and high SES. The adjusted *p*-value indicates a true association between the consumption of sweets and middle SES (0.0067, 95% CI) and affluent SES (0.000, 95% CI).

Table 5 below indicates the post hoc test for lunchbox contents vs. school-level SES the adjusted *p*-value indicates snack association between the consumption of snacks and low socioeconomic status (*p* = 0.0014, 95% CI). There is a true association between junk food among middle SES (*p* = 0.0061, 95% CI) and affluent SES (*p* = 0.00693, 95% CI). The adjusted *p*-value indicates that there is a true association between the consumption of fruits in terms of SES, there is high consumption among affluent SES (*p* = 0.000, 95% CI) and low consumption among low SES (*p* = 0.001, 95% CI). Also, the adjusted *p*-value indicates that there is a true association between vegetable consumption in terms of SES, there is high consumption among affluent SES (*p* = 0.000, 95% CI) and low consumption among the low SES (*p* = 0.000, 95% CI). The adjusted *p*-value indicates that there is a true association between sweetened beverage consumption among the low SES (0.0019, 95% CI) but not for the middle and high SES. The adjusted *p*-value indicates a true association between the consumption of sweets among middle SES (0.0067, 95% CI) and affluent SES (0.000, 95% CI).

## 4. Discussion

This study aimed to scan for the obesogenicity of primary school environments in Tshwane. This study hypothesised that Tshwane primary school environments are obesogenic. This study revealed that the hypothesis is correct. Similar findings have been reported in both developed and developing countries. A recent cross-sectional study was conducted among Brazilian schoolchildren to assess their school environments. It indicated that there was an association between obesity and the availability of ready-to-eat food shops around the school [21]. Vega-Salas et al. conducted a systematic review of interventions and policies relating to schoolchildren in Latin America and the Caribbean. The findings of the systematic review recommended that to prevent childhood obesity, schools needed to implement nutrition and physical education initiatives [22]. A study conducted in California, USA, found that risk factors for childhood obesity included the availability of vending machines around schools, access to SSBs, lack of physical education, and nonparticipation in physical activities. The lack of restricting policies on nutrition was also cited.

This study found that most schools have a nutrition policy. Most public schools had policies on nutrition and physical education, even dedicated classes on both, which means that they comply with the National School Nutrition Programme (NSNP) [23]. Only one in ten (10%) of the primary schools reported not having these policies, which places them in violation of the South African Basic Education Act. Two-thirds of the schools reported that these policies had not been communicated to staff, meaning that they had merely been received and stored on school premises. This is evidenced by the poor implementation of nutrition policies, i.e., the absence of nutrition and physical activity posters in classrooms and the absence of vegetable gardens on school premises. Although South Africa is characterised by many disparities, as witnessed in many of the underprivileged schools studied, it is commendable that policies regarding nutrition and physical education are available to all schools. However, failure to communicate these policies has led to poor compliance regarding the implementation of these policies in schools.

A study in Australia found that most schools did not have a nutrition policy, and only three of the seven schools managed to implement healthy eating practices [24]. The main barriers to implementing nutrition policies were the nonprioritisation of nutrition policies by school managers, and poor support from parents, staff, stakeholders, and schoolchildren themselves [24]. In the Philippines, a qualitative study was conducted among policymakers on the regulation of a school feeding policy. The policymakers indicated that it was difficult for them to regulate school feeding policies due to a lack of human and financial resources, the wording of the policies not being clear, and the existing relationships between food companies and schools [25].

The researchers in this study found that most schools did not comply with most of the items stipulated by the DBE, making the schools obesogenic environments. Most of the schools included in this study did not have educational materials on nutrition and exercises, the schoolchildren did not engage in exercises, the teachers were not knowledgeable about exercises, and the tuck shops did not comply with the DBE Guidelines for Tuck Shop Operators. In Malaysia, a school environmental mapping framework based on the ANGELO framework was conducted [26]. Most Malaysian schools adhered to the entire environmental mapping framework, especially in terms of the physical environment. In Malaysia, the physical environment included curricular and physical support, food and drink provision, food and drinks in the school canteen, and health, nutrition, and physical activity programmes. Urban schools complied more than rural schools.

This study showed that Gauteng schools did not comply with the environmental economic mapping. School tuck shops were mainly selling unhealthy food items such as SSBs, snacks, sweets, and junk food. In contravention of the DBE Guidelines for Tuck Shop Operators, tuck shops did not sell healthy foods such as fruits and vegetables. Similar findings have been reported in the Netherlands. In the urban Netherlands, a study by Timmermans et al. [27] revealed that there were many retail outlets surrounding schools. Most of those outlets sold sugar-containing drinks, fries, and hamburgers, with very few selling fruits (23%) and vegetables (12.2%). According to the DBE Guidelines for Tuck Shop Operators, schools must sell healthy food items such as fruits and unsalted peanuts. The sale of healthy options will probably increase the consumption of healthy foods among schoolchildren. Rossettie et al. [28] studied the impact of the national implementation of policies regarding the provision of fruit and vegetables as well as restrictions on sugar-sweetened beverages (SSBs) in the United States. The survey revealed that the implementation of national school policies on the provision of fruits and vegetables would likely improve consumption, while SSB restrictions reduce consumption, thus preventing the consumption of obesogenic foods.

This study revealed that there was an association in terms of fruit and vegetable consumption, i.e., affluent were consuming more fruits and vegetables compared to low-SES and middle-SES children. It also revealed an association between low SES and snack consumption. There was a true association between low consumption of junk food and middle SES and affluent SES, as compared to its low consumption among low-SES students. The post hoc test indicated an association between high consumption of sweetened beverages and low SES. From these findings, we can deduce that low SES is associated with high consumption of junk, SSBs, and snacks, and low consumption of fruits and vegetables.

A study conducted in Nepalese schools indicated that learners in public schools ate more junk food compared to learners in private schools. It also indicated a more frequent consumption of fruits and vegetables among learners in private schools compared to public school learners. In countries with low socioeconomic status, such as Thailand, the supply of fruits and vegetables is limited [29]. Studies, including this one, show that unhealthy eating is common among learners, especially in areas of low socioeconomic status. This could be alleviated by providing fruits and vegetables in the school feeding programme, as most of them consume food from the programme.

This study showed that despite the existence of dedicated sports grounds, schoolchildren were not participating in sports or exercises. This may have been due to this study being conducted during the COVID-19 lockdown. However, the literature shows that lockdown regulations aside, there are other barriers preventing schoolchildren from exercising. Somerset and Hoare [30] conducted a systematic review among children in different settings in Europe. They found that the main reasons for not participating in sports were time and cost. Time included a tight schedule such as homework, not being able to take a child to a sports venue, and transportation to and from sports venues. Costs included sports resources, such as affiliation fees and equipment, and money to pay for transportation. 

This study also indicated that very few teachers knew about exercises and exercise activities according to the DBE’s ATP could have sparked the lack of interest in leading children to exercises. Furthermore, schools substituted physical activity periods with other academic classes. This practice served as a barrier to schoolchildren’s participation in exercises. Findings by Pirzadeh et al. indicate that teachers lack knowledge about nutrition and exercise as they have never received formal education about these subjects [31]. A systematic review conducted by Duffey et al. [32] among European adolescent girls revealed that barriers to participating in sports were a lack of support from peers, family, and teachers, as well as a lack of time.

Although COVID-19 restrictions may have negatively impacted participation in exercises, the regulations were relaxed during the data collection. Exercises could have easily been performed in smaller groups, with restrictions adhered to. Exercise is an important measure to prevent a sedentary lifestyle, which ultimately plays a role in obesity prevention. Exercise is important for health and weight control. Exercise activities may include walking to school, playing during breaks, and reducing screen time [33]. The WHO global strategy (WHO, 2018) recommends that children between the ages of 5 and 17 years should engage in at least 60 min of moderate-to-vigorous-intensity physical activity daily [34]. To improve adherence at schools, the DBE should pay attention to the lack of exercise knowledge amongst teachers and hire exercise experts for schools, which is already effective in some quintile five schools in Tshwane. A qualitative cross-sectional and descriptive survey recently conducted in South Africa highlighted that challenges with implementing SAFBDGs in primary schools include a lack of nutrition and exercise knowledge, time constraints to teach SAFBDGs, teachers’ unhealthy lifestyles and high BMIs, and a lack of resources such as posters and pamphlets to teach with [35]. Armstrong et al. (2011) highlighted the need for formal physical education in schools [36].

This study shows that current initiatives and facilities in aid of nutrition and physical activity are insufficient and ineffective in improving nutrition and exercise practices among primary schoolchildren. The Department of Basic Education (DBE) needs to collaborate with other sectors such as the Nutrition Directorate to educate children about the benefits of exercise and healthy eating. The DBE has a responsibility to provide schools with policy documents, to ensure that these policies are communicated to educators, and to monitor compliance in the implementation of these policies.

Although tuck shops on school premises do not sell healthy foods to children, as revealed in our study, the DBE’s Guidelines for Tuck Shop Operators should be communicated to school governing bodies (SGBs), all staff in schools, parents, street vendors, and tuck shop owners.

School authorities have a responsibility to monitor what is sold by both street vendors and tuck shops, and to ensure compliance with the Guidelines for Tuck Shop Operators as stipulated in the DBE policy document. The DBE should consider hiring full-time nutrition and physical education experts per district. These experts would oversee nutrition education, which would assist in changing the obesogenicity of the school environment and changing school culture to promote healthy eating.

### 4.1. Study Limitations

Data collection was conducted from April to July 2021. It was during COVID-19 lockdown, for which reason some schools were very strict and could not grant access to external visitors. This affected the selection of some schools, even though they were eligible. Furthermore, restrictions affected exercise participation and schoolchildren were attending school on a rotational basis.

Furthermore, this study was carried out in only ten public schools. It is recommended that this study be conducted in all schools in the Tshwane West district of Gauteng, so that customised interventions can be implemented as required per school. However, the ANGELO framework used in the analysis of the obesogenicity of primary school environments made this study unique, as it had never been used before in South Africa before.

### 4.2. Practical Implications

The findings of this study indicate that school environments in Tshwane are obesogenic. These findings imply that existing education and country-level efforts to prevent childhood obesity need to be improved and aided by the scanning and improving school environments. The results imply that there is a challenge in communicating and monitoring policies at the school level, coupled with a lack of depth in delivering nutrition and physical activity educational information in primary schools. There is a need for the Department of Education to formally reskill Life Skills and Life Orientation educators so that they have more impact on children.

The existing Guidelines for Tuck Shop Operators must be communicated to the end-users. Street vendors and owners of tuck shops surrounding schools must be trained and orientated on these guidelines to comply. School environments must be converted into health-promoting environments which champion health for all.

## 5. Conclusions

This study contributes to the body of knowledge, based on the findings that primary school environments are obesogenic and therefore enable of childhood obesity. It is essential to perform a wider observational study to understand to what extent school environments promote obesogenicity in all schools in South Africa so that tailor-made interventions can be designed and implemented for the prevention of childhood obesity. Multisectoral collaboration is needed in the development and implementation of nutrition and exercise interventions to prevent childhood obesity in South Africa.

## Figures and Tables

**Figure 1 ijerph-20-06889-f001:**
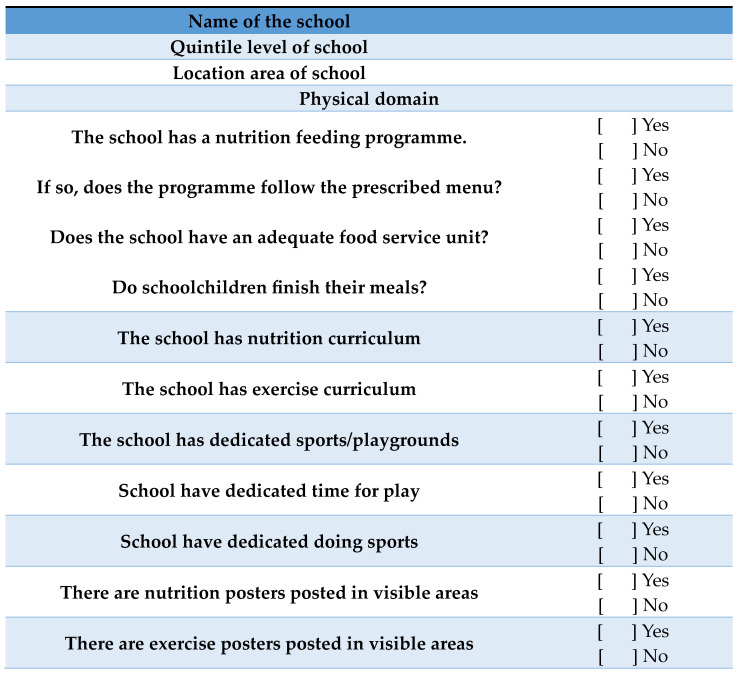
Observation checklist.

**Figure 2 ijerph-20-06889-f002:**
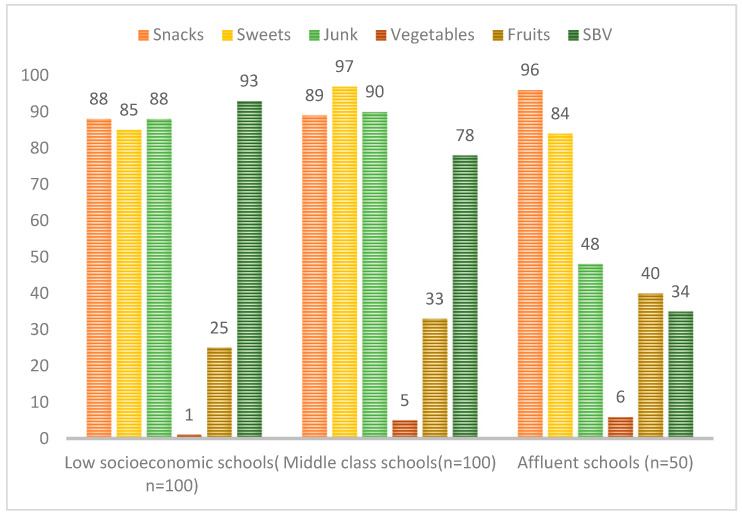
The total number (*n* = 250) of lunchboxes inspected according to the contents of the lunchboxes and socioeconomic status.

**Table 1 ijerph-20-06889-t001:** The frequency distribution of the nutrition domains.

Variable (N = 10)	*n* (%)
The school has a nutrition policy	9 (90%)
The nutrition policy is communicated to all staff	4 (40%)
The school has nutrition posters or pamphlets	2 (20%)
The school has street vendors	10 (100%)
Nutrition education is part of the curriculum	10 (100%)
The school has a tuck shop	5 (50%)
The school has implemented DBE Guidelines for Tuck Shop Operators	0 (0%)
The school has a copy of the South African Food-Based Dietary Guidelines (SAFBGs)	1 (10%)
The school has a garden infrastructure	2 (20%)
The school has a functional garden	0 (0%)

**Table 2 ijerph-20-06889-t002:** The frequency distribution of exercise domains.

Variable (N = 10)	*n* (%)
The school has dedicated sports grounds	10 (100%)
Primary schoolchildren engage in exercises	0 (0%)
Exercise education is part of the curriculum	10 (100%)
Exercise posters or pamphlets are available	0 (0%)
The school is fully equipped with exercise equipment	5 (50%)
Teachers have adequate knowledge regarding exercises	1 (10%)
Teachers can design exercises according to the DBE Annual Teaching Plan (ATP)	1 (10%)

**Table 3 ijerph-20-06889-t003:** Scanning of the school environment for the availability of listed items.

Variable (N = 10)	*n* (%)
The school has a feeding programme	9 (90%)
The school follows the prescribed menu	5 (50%)
The school has an adequate FSU	5 (50%)
The schoolchildren finish their food	8 (80%)

**Table 4 ijerph-20-06889-t004:** Crosstabulation of contents of lunchboxes vs. school SES level.

	Socioeconomic Status of the School			
Contents of Lunchbox	Low Socioeconomic School (*n* = 100)	Middle Socioeconomic School (*n* = 100)	Affluent Socioeconomic School (*n* = 50)	*p*-Value and χ^2^
Snacks	88 (88%)	89 (89%)	48 (96%)	*p* = 0.279 χ^2^ = 2.556
Junk	87 (87%)	89 (89%)	24 (48%)	*p* < 0.001 * χ^2^ = 40.125
Fruits	25 (25%)	33 (33%)	20 (40%)	*p* = 0.154 χ^2^ = 3.746
Vegetables	1 (1%)	5 (5%)	3 (6%)	*p* = 0.188 χ^2^ = 3.343
Sugar-sweetened beverages	93 (93%)	79 (79%)	17 (34%)	*p* < 0.001 * χ^2^ = 63.947
Sweets	85 (85%)	97 (97%)	42 (88%)	*p* = 0.007 * χ^2^ = 9.830

* Indicates significance.

**Table 5 ijerph-20-06889-t005:** Post hoc test for contents of lunchboxes vs. school SES level.

Contents of Lunchbox	Adjusted Residual Adjusted Chi-Squared Adjusted *p*-Value	Socioeconomic Status of the School		
		Low Socioeconomic School (*n* = 100)	Middle Socioeconomic School (*n* = 100)	Affluent Socioeconomic School (*n* = 50)
Snacks	Adjusted residual Adjusted *p*-value Adjusted chi-squared	3.8 *p* = 0.0014 * χ^2^ = 14.44	2.2 *p* =0.2781 χ^2^ = 4.84	2.1 *p* = 0.3573 χ^2^ = 4.41
Junk	Adjusted residual Adjusted *p*-value Adjusted chi-squared	1.4 *p* = 0.1615 χ^2^ = 3.61	2.7 *p* = 0.0061 * χ^2^ = 11.56	5.0 *p* = 0.00693 * χ^2^ = 42.25
Fruits	Adjusted residual Adjusted *p*-value Adjusted chi-squared	4.8 *p* = 0.001 * χ^2^ = 24.01	0.7 *p* = 0.4830 χ^2^ = 0.49	6.7 *p* = 0.000 * χ^2^ = 44.89
Vegetables	Adjusted residual Adjusted *p*-value Adjusted chi-squared	4.9 *p* = 0.000 * χ^2^ = 24,01	1.8 *p* = 0.7186 χ^2^ = 3.24	8.2 *p* = 0.000 * χ^2^ = 67.24
Sugar-sweetened beverages	Adjusted residual Adjusted chi-squared Adjusted chi-squared	3.1 *p* = 0.0019 * χ^2^ = 9.61	2.5 *p* = 0.1242 χ^2^ = 6.25	0.8 *p* = 0.4237 χ^2^ = 0.64
Sweets	Adjusted residual Adjusted *p*-value Adjusted chi-squared	1.9 *p* = 0.5743 χ^2^ = 3.61	3.4 *p* = 0.0067 * χ^2^ = 11.56	6.5 *p* = 0.000 * χ^2^ = 42.25

* Indicates significance.

## Data Availability

Data are available on request from the corresponding author.

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
