# Peer review of "Scanning for Obesogenicity of Primary School Environments in Tshwane, Gauteng, South Africa"

_ijerph, 2023, doi:10.3390/ijerph20196889_

Round 1

Reviewer 1 Report

The aim of this observational study was to assess the obesogenic factors in primary school environments in a sample of schools in South Africa. The topic of the manuscript is important for public health. There are my comments that might help to increase the quality of this manuscript:

Abstract

1.       Please delete the sentence in lines 13-14.

2.       Please provide more specific results in the abstract.

 Introduction

1.       The Introduction covers the topic; however, it might be more focused.

2.       More research that used a similar observational methodology might be introduced and discussed.

3.       The authors mention sedentary behaviour as one of the factors of the obesogenic environment. However, sedentary behaviour is not introduced or discussed further.

4.       The authors use abbreviations without explaining them first (i.e., SA, SAFBDGs). The aim of the study

5.       The aim of the study was “to investigate to what extent school environments promote obesogenicity” in primary schools in South Africa. First, the study does not represent all of Africa, therefore more tempered language is recommended here. Second, the authors did not know before implementing the study if obesogenic environments are prevalent in a particular sample of primary schools. Please, revise the aim of the study. Please also develop hypotheses for the study.

6.       Referencing all documents should be provided, the same is for Discussion.

 Methods

1.       Methods lack procedure, please describe how the observation of school was implemented.

2.       Ethical considerations are missing.

3.       The number of observed schools is unclear.

4.       Referencing for the ANGELO framework should be provided. Was sedentary behaviour observation part of the ANGELO?

5.       It is unclear how schools were divided into different categories of socioeconomic level.

6.       How do researchers divide lunch boxes if vegetables and junk foods were found together?

7.       How was the knowledge of teachers about exercises assessed in this study?

Results

1.       The colours in the figures should be changed since they are similar in a white and black version of the article, also I recommend revising figure 1 so that only yes or no answers might be seen. No name of the figure might be presented in the figure itself. The name of Figure 1 should be revised without commenting on the data. The same is for Fig. 2.

Discussion

1.       Please shortly describe what was the main aim and the assumptions of the present study and then go to the main findings and discuss what novel findings globally the study adds. Then discuss other findings.

2.       The subsection “Practical Implications” might be included including paragraphs that relate to the topic.

Conclusions

1.       The conclusions should be more specific and provide clear insight into what was found.

The English language is problematic in this manuscript, professional proofreading is recommended.

Author Response

Dear reviewer 

Regards

Reviewer 2 Report

Here are my comments and suggestions:

1-Line 41, please indicate the other contributing factors to obesity among primary school children.

2-Line 45 high unsaturated fat? Reference 11 does not specify what type of fat (unsaturated or saturated), so please indicate only fat.

3-line 49, Acronym SA, please specify the entire word, maybe South Africa?

4-LIne 67, there is a typo, the word schools is mentioned twice.

5-line 70, please specify the entire word of SAFBDG.

6-Lines, 102-114, Consider using a table to indicate the checklist used in this study, in this way, the reading is more fluent.

7-Lines 115-119 indicate the software used for the data analysis.

8-The references 1 and 2 are poorly written, please insert the correct reference. Organization WH? Maybe is WHO.

In addition, I would to know why is not there any statistical correlation between the data present in the result?

Explicit the motivation (e.g. paucity of data or other) of why the statistical analysis is not present, or consider inserted.

none

Author Response

Dear reviewer

Regards

Round 2

Reviewer 1 Report

 The authors addressed the majority of my comments. There are some additional comments:

Mistakes (typos) in lines 32, 42, 70. All text should be checked for it.

In the 6th paragraph (lines 73-76) authors state ...“despite evidence showing that sport and physical activity could be effective in promoting exercise“. This part of the sentence is confusing, please revise it.

The aim of the study should be clear. Please write: „The aim of the study was to .... „  Then present the hypothesis (lines 88-93).

Do the authors have permission to reprint the schematic diagram in Fig. 1.?  Please double-check this point.

The information from Table 3 might be presented in the text. It is no need to present such a big table to present several numbers.

The percentage in Fig. 1 is 120, please revise it. I recommend presenting the data from Fig. 1 in Table format.

Author Response

Dear Editor 

Regards

Reviewer 2 Report

The authors have responded to all of reviewer's suggestions. In this way the paper is ready to be published if the editor agree this.

None

Author Response

Dear Editor

Thanks for reviewing and approving my manuscript. 

Regards